# First mobilisation after abdominal and cardiothoracic surgery: when is it actually performed? A national, multicentre, cross-sectional study

Monika Fagevik Olsén [1], Maria Sehlin,[2] Elisabeth Westerdahl,[3] Anna Schandl,[4] Linda Block,[5] Malin Nygren-Bonnier,[6] Anna Svensson-Raskh [6]

**Correspondence to**
Dr Monika Fagevik Olsén;
monika.fagevik-olsen@gu.se

## ABSTRACT

**Objectives** Knowledge of clinical practice regarding mobilisation after surgery is lacking. This study therefore aimed to reveal current mobilisation routines after abdominal and cardiothoracic surgery and to identify factors associated with mobilisation within 6 hours postoperatively.

**Design** A prospective observational national multicentre study.

**Setting** 18 different hospitals in Sweden.

**Participants** 1492 adult patients undergoing abdominal and cardiothoracic surgery with duration of anaesthesia >2 hours.

**Primary and secondary outcomes** Primary outcome was time to first postoperative mobilisation. Secondary outcomes were the type and duration of the first mobilisation. Data were analysed using multivariate logistic regression and general structural equation modelling, and data are presented as ORs with 95% CIs.

**Results** Among the included patients, 52% were mobilised to at least sitting on the edge of the bed within 6 hours, 70% within 12 hours and 96% within 24 hours. Besides sitting on the edge of the bed, 76% stood up by the bed and 22% were walking away from the bedside the first time they were mobilised. Patients undergoing major upper abdominal surgery required the longest time before mobilisation with an average time of 11 hours post surgery. Factors associated with increased likelihood of mobilisation within 6 hours of surgery were daytime arrival at the postoperative recovery unit (OR: 5.13, 95% CI: 2.16 to 12.18), anaesthesia <4 hours (OR: 1.68, 95% CI: 1.17 to 2.40) and American Society of Anaesthesiologists (ASA) classification 1–2, (OR: 1.63, 95% CI: 1.13 to 2.36).

**Conclusions** In total, 96% if the patients were mobilised within 24 hours after surgery and 52% within 6 hours. Daytime arrival at the postoperative recovery unit, low ASA classification and shorter duration of anaesthesia were associated with a shorter time to mobilisation.

**Trial registration number** FoU, Forskning och Utveckling in VGR, Vastra Gotaland Region (Id:275357) and Clinical Trials (NCT04729634).

## STRENGTHS AND LIMITATIONS OF THIS STUDY

⇒ The prospective, multicentre design of this study, including patients from all parts of Sweden, and the relatively high participation rate counteract selection bias.

⇒ The large study sample enabled subgroup analyses between the different types of surgery and exploration of related variables.

⇒ Outcomes were determined by observations and relied on correct documentation by investigators.

⇒ Patients were not recruited during weekends, and therefore the results may not completely reflect common practice.

for postoperative recovery and prevention of complications, even though its specific effects are largely unexplored.[1 2] During decades of development of pre, peri and postoperative care, it has been found that other active interventions such as reinflation of the lungs at the end of anaesthesia may reduce the risk of postoperative pulmonary complications.[3–6] However, the specific effects of mobilisation are not yet known, nor it is clear how to facilitate it or when it should be initiated.[7 8]

Undoubtedly, the change of position to sitting and/or standing has a multifactorial positive impact on cardiorespiratory function.[8–10] Lung volume increases with a subsequent decrease of the atelectic area, improving the ventilation/perfusion ratio and facilitating the removal of secretions.[11–13] The immediate respiratory effects of mobilisation have recently been evaluated in two randomised, controlled trials.[14 15] In those, mobilisation within the first hours after surgery was compared with immobilised control groups. The results indicate that changing position from supine to upright has advantages for cardiorespiratory function and no major adverse events of the intervention were reported.

## BACKGROUND

Mobilisation after abdominal and cardiothoracic surgery is an important intervention

Most hospitals claim that 'early' mobilisation is implemented in their standard practice of care, but the term 'early' is not clearly defined,[7] and the current state of practice remains unknown. Therefore, the present study aimed to estimate the time of first mobilisation and to identify factors associated with a mobilisation within 6 hours in a large cohort of adult patients who had undergone acute, subacute or elective abdominal or cardiothoracic surgery.

## MATERIALS AND METHODS

### Design and setting
This is a part of the prospective observational national multicentre cohort study entitled 'Survey of mobilisation and breathing exercise after cardiothoracic and abdominal surgery'. A total of 18 hospitals representing all parts of Sweden participated, including 7 university hospitals and 11 county or local hospitals, (see online supplemental table 1). Data was collected between September 2021 and February 2022. The STrengthening the Reporting of OBservational studies in Epidemiology checklist (STROBE) was used to report this data.

### Patient and public involvement
Patients or public were not involved in the design, or conduct, or reporting, or dissemination plans of our research.

### Context of the study
In Sweden, most patients who have undergone abdominal and cardiothoracic surgery are treated and monitored at intensive care units (ICUs), specific postoperative units or intermediate units (a lower level of care than ICU, but a higher level of care than surgical wards), for some hours until they are considered to have sufficient circulatory and respiratory stability to be transferred to surgical wards. The standard care includes mobilisation within the first day after surgery, assisted by nurses, assistant nurses and physiotherapists. Some hospitals included in this study followed multimodal perioperative care pathways for some of the categories of patients.

### Inclusion
Adult patients (≥18 years) who had undergone open, laparoscopic or robot-assisted surgery within the thorax or abdomen, with a duration of anaesthesia exceeding 2 hours, were eligible for inclusion. Patients who had undergone plastic, trauma, transplant or orthopaedic surgery or reoperation were excluded. Data was captured on patients undergoing surgery from Monday to Thursday.

At each study site, one designated staff member (a physiotherapist or a nurse experienced in surgical practice) was responsible for inclusion screening of all patients and for controlling that all data were collected.

### Data collection
Patient characteristics and preoperative variables were retrieved from the patients' records. Data on perioperative variables, such as surgical procedure, duration of anaesthesia (in hours and minutes), blood loss (mL), type of anaesthesia and functional status (partly/totally dependent) were obtained from the operation and postoperative surveillance charts. Type of surgery was categorised into: cardiac (coronary artery bypass grafting, valve or aortic surgery via sternotomy); thoracic (pulmonary surgery via thoracotomy or thoracoscopy); major upper abdominal (pancreatic, hepatic, oesophageal and ventricular open procedures); minor upper abdominal (open and laparoscopic procedures such as fundoplication and cholecystectomy); intestinal (smaller intestine surgery and/or colorectal surgery); major lower abdominal (extensive, open, gynaecological, urological and other procedures with incision below the umbilicus); minor lower abdominal (minor and laparoscopic gynaecological, urological and other procedures with incision below the umbilicus).

The original research protocols were identifiable during the collection process to enable data completion from the patients' records. When data was complete, the protocols were deidentified before entering the results into a secure online database.

### Outcomes
The primary outcome was time to first mobilisation from arrival at the postoperative unit. Mobilisation was defined as being able to at least sit on the edge of the bed. Secondary outcomes were type and duration of the first mobilisation and number of staff assisting in mobilisation. Based on the literature and clinical reasoning, the following data were considered potential factors influencing an early mobilisation (defined as within 6 hours of surgery) and were therefore collected; age, body mass index (BMI), smoking, duration of anaesthesia, perioperative bleeding, comorbidity (ASA classification), type of surgery and arrival at the postoperative recovery unit.[3 16–18]

The outcome data were registered, in parallel with the clinical work, in a case report form following the patient until 24 hours from admission to a postoperative unit. If no mobilisation was performed during the first 24 hours, the reason was noted in a bedside chart by the staff responsible for the trial at each study site. If any of the variables had been missed to be filled in a protocol, these specific patient's results were excluded from the analysis.

### Statistical analysis
Continuous data were presented as means (SD) or median (IQR), while categorical data were illustrated with numbers and percentages (%). Differences between groups (types of surgery) were analysed with Analysis of Variance (ANOVA), Kruskal-Wallis test or $\chi^2$ test.

The impact of baseline variables on the proportion of subjects and time to first mobilisation, time out of bed and type of mobilisation respectively, was analysed with univariable logistic regression. The significant univariable predictors were then entered stepwise into a multivariable

logistic regression, using forward selection. Data were presented as ORs with a 95% CI and p value.

A general structural equation model (GSEM) was applied to identify the associations between potential factors for being mobilised early and the time to first mobilisation, defined as within 6 hours from arrival at the postoperative unit.[19] A univariate logistic model identified relevant factors: age, sex, BMI, perioperative bleeding, smoking, duration of anaesthesia, ASA classification, time of arrival at the postoperative recovery unit and type of surgery. Following this, a multivariate logistic regression was performed (including variables based on the statistical findings p<0.05), and to test for collinearity. The included variables were assessed stepwise and classified into the following categories: ASA classification (1–2 and 3–4)[11 17]; age (≤70 and >70 years),[3 11 18] arrival at the postoperative recovery unit (daytime/afternoon and evening/night-time)[11]; type of surgery (intestinal, minor upper and lower abdominal vs cardiac, thoracic, major upper and lower abdominal)[11 18]; and duration of anaesthesia (<4 and ≥4 hours).[3 11 18] Finally, a GSEM was applied

to analyse the structural association between the variables and mobilisation within 6 hours of arrival at the postoperative unit.

IBM SPSS V.29.0 and Stata Statistical Software (Release V.17, College Station, Texas, USA, Stata Corp LLC) were used for the analyses.

## RESULTS

Among 3802 patients who underwent abdominal or cardiothoracic surgery at the hospitals during the study periods, 1879 met the inclusion but not the exclusion criteria. Among them, 1492 patients (79%) were included in the study (figure 1 and table 1).

### Time to first mobilisation

Among the included patients, 279 (20%) were mobilised within 3 hours, 738 (52%) within 6 hours, 997 (70%) within 12 hours and 1396 (96%) within 24 hours. The mean time to the first mobilisation was 7 hours 51 min (SD 6:27) (table 2). Major upper abdominal surgery was

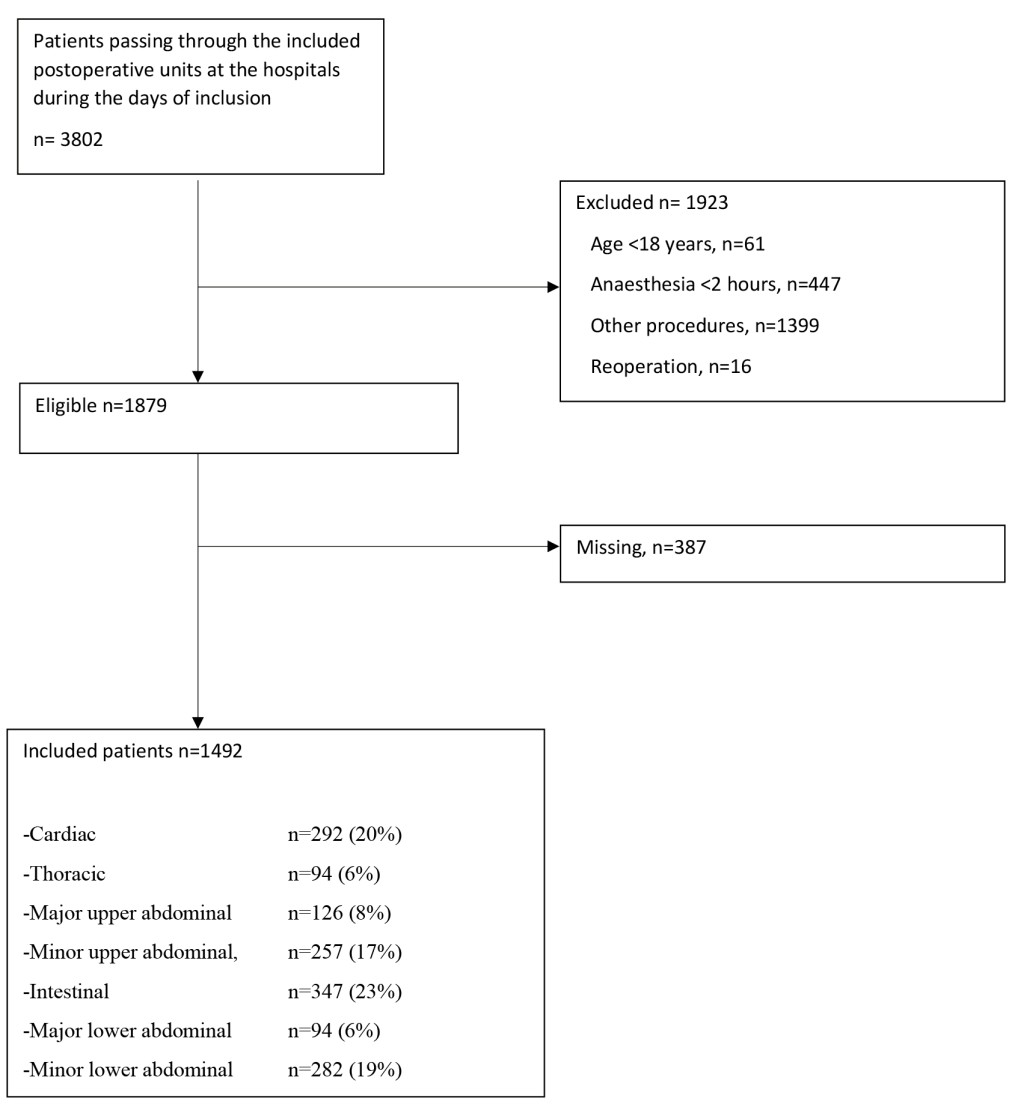

**Figure 1** Flow chart of the patients included in the study.

**Table 1** Demographic data of the included patients

| | All, n=1492 | Cardiac, n=292 | Thoracic, n=94 | Major upper Abd, n=126 | Minor upper Abd, n=257 | Intestinal n=347 | Major lower Abd n=94 | Minor lower Abd n=282 |
|---|---|---|---|---|---|---|---|---|
| **Preoperative factors** | | | | | | | | |
| Age, years | 62 (16) | 64 (14) | 60 (18) | 67 (13) | 56 (18) | 67 (16) | 62 (14) | 60 (15) |
| Male sex, n | 784 (49.1%) | 227 (77.7%) | 49 (52.1%) | 81 (64.3%) | 96 (37.4%) | 175 (50.4%) | 33 (35.1%) | 123 (43.6%) |
| Body mass index, kg/m$^2$ | 27.7 (5.5) | 27.0 (4.9) | 25.7 (4.5) | 27.0 (4.8) | 29.1 (6.4) | 26.3 (5.8) | 26.4 (5.1) | 27.0 (5.3) |
| Functional status—partly/totally dependent, n | 18 (1.1%) | 2 (0.7%) | 0 | 1 (0.8%) | 5 (1.9%) | 9 (2.6%) | 0 | 1 (0.4%) |
| **ASA, Score** | | | | | | | | |
| 1 | 171 (11.8%) | 0 | 6 (6.4%) | 8 (6.8%) | 42 (17.1%) | 30 (8.8%) | 14 (14.9%) | 71 (25.5%) |
| 2 | 575 (39.5%) | 12 (4.2%) | 37 (39.4%) | 53 (44.9%) | 119 (48.4%) | 156 (46.0%) | 64 (68.1%) | 134 (48.2%) |
| 3 | 611 (42.0%) | 212 (74.4%) | 48 (51.1%) | 56 (47.5%) | 73 (28.4%) | 136 (40.1%) | 15 (16.0%) | 71 (25.5%) |
| 4 | 96 (6.6%) | 60 (21.1%) | 3 (3.2%) | 1 (0.8%) | 12 (4.7%) | 17 (5.0%) | 1 (1.1%) | 2 (0.7%) |
| 5 | 1 (0.1%) | 1 (0.4%) | 0 | 0 | 0 | 0 | 0 | 0 |
| Missing | 38 | 7 | 0 | 8 | 11 | 8 | 0 | 4 |
| **Perioperative factors** | | | | | | | | |
| Duration of anaesthesia, hours | 4.60 (2.17) | 4.80 (1.46) | 3.37 (1.12) | 7.22 (3.12) | 3.43 (1.35) | 5.26 (2.45) | 4.72 (1.81) | 3.77 (1.41) |
| Duration of surgery, hours | 3.27 (2.16) | 3.41 (1.95) | 2.03 (1.07) | 5.53 (2.98) | 2.33 (1.20) | 3.90 (2.29) | 3.34 (1.73) | 2.59 (1.74) |
| Open surgery, n | 826 (55.6%) | 282 (96.9%) | 31 (33.0%) | 102 (81.6%) | 68 (26.7%) | 195 (53.5%) | 60 (63.8%) | 88 (31.3%) |

Data presented as mean (SD) or n (%).
Abd, abdominal; ASA, American Society of Anaesthesiologists classification.

**Table 2** Time to first mobilisation, duration and type of mobilisation, presented as mean (SD), median (Q1:Q3) or n (%)

| | All, n=1492 | Cardiac, n=292 | Thoracic, n=94 | Major upper Abd, n=126 | Minor upper Abd, n=257 | Intestinal, n=347 | Major lower Abd, n=94 | Minor lower Abd, n=282 |
|---|---|---|---|---|---|---|---|---|
| Time to first mobilisation, hours* | 7.91 (5.93) 5.50 (3.30:13.02) | 8.20 (5.47) 6.67 (4.00:12.50) | 8.39 (6.61) 4.92 (3.31:15.06) | 11.05 (5.80) 11.80 (4.84:16.29) | 5.95 (5.63) 3.75 (2.22:7.41) | 7.48 (8.02) 5.25 (3.25:13.72) | 8.75 (6.21) 6.10 (3.31:15.70) | 7.63 (5.68) 2.26 (3.92:8.71) |
| Time to first mobilisation, categorised in intervals, hours* | | | | | | | | |
| <3 | 279 (20.4%) | 56 (20.1%) | 15 (16.9%) | 11 (9.3%) | 83 (36.6%) | 67 (21.6%) | 15 (17.0%) | 32 (12.4%) |
| 3 to <6 | 458 (33.4%) | 64 (22.9%) | 39 (44.0%) | 25 (21.2%) | 76 (33.5%) | 111 (35.8%) | 28 (31.8%) | 115 (44.4%) |
| 6 to <9 | 186 (13.6%) | 55 (19.7%) | 4 (4.5%) | 9 (7.6%) | 18 (7.9%) | 36 (11.6%) | 14 (15.9%) | 50 (19.3%) |
| 9 to <12 | 74 (5.4%) | 29 (10.4%) | 5 (5.6%) | 12 (10.2%) | 9 (4.0%) | 15 (4.8%) | 2 (2.2%) | 2 (0.8%) |
| 12 to <15 | 103 (7.5%) | 30 (10.8%) | 8 (9.0%) | 20 (16.9%) | 11 (4.8%) | 18 (5.8%) | 3 (3.4%) | 13 (5.0%) |
| 15 to <18 | 160 (11.7%) | 30 (10.8%) | 10 (11.2%) | 28 (23.9%) | 15 (6.6%) | 35 (11.3%) | 18 (20.5%) | 24 (9.3%) |
| 18 to <21 | 87 (6.4%) | 15 (5.4%) | 3 (3.4%) | 11 (9.3%) | 12 (5.3%) | 22 (7.1%) | 7 (8.0%) | 17 (6.6%) |
| 21–24 | 23 (1.8%) | 0 | 5 (5.6%) | 2 (1.8%) | 3 (1.3%) | 6 (1.8%) | 1 | 6 (2.3%) |
| No time reg. | 26 | 0 | 2 | 2 | 6 | 7 | 3 | 6 |
| Not mobilised | 62 | 13 | 2 | 5 | 15 | 22 | 1 | 4 |
| Missing | 34 | 0 | 1 | 1 | 9 | 8 | 2 | 13 |
| Mobilised within 24 hours, n | 1430 (95.8%) | 279 (95.9%) | 91 (97.8%) | 120 (96.0%) | 233 (94.0%) | 317 (94.3%) | 91 (98.9%) | 265 (98.5%) |
| Duration of mobilisation, hours* | 0.45 (1.27) 0.17 (0.17:0.33) | 0.34 (0.68) 1.67 (0.08:0.25) | 0.96 (2.55) 0.50 (0.17:1.00) | 0.49 (0.69) 0.25 (0.17:0.50) | 0.58 (1.82) 0.25 (0.17:0.48) | 0.31 (0.41) 0.67 (0.67:0.33) | 0.27 (0.29) 0.67 (0.12:0.30) | 0.47 (1.56) 0.17 (0.08:0.33) |
| Sitting on the edge of the bed | 1395 (100) | 279 (100) | 91 (100) | 119 (100) | 234 (100) | 317 (100) | 91 (100) | 265 (100) |
| Standing by the bed* | 1059 (75.9) | 248 (88.9) | 75 (82.4) | 85 (71.4) | 182 (70.8) | 197 (62.1) | 73 (80.2) | 199 (75.4) |
| Sitting in a chair* | 203 (14.6) | 42 (15.1) | 47 (51.6) | 31 (26.1) | 23 (9.8) | 31 (8.9) | 10 (11.0) | 19 (7.2) |
| Walking* | 294 (21.1) | 1 (0.4) | 17 (18.7) | 8 (6.7) | 112 (47.9) | 42 (13.2) | 19 (20.9) | 95 (36.1) |
| Distance walked, m* | 8.3 (5.8) | 7.5 (10.6) | 45.9 (46.8) | 33.3 (32.7) | 23.0 (26.3) | 32.3 (32.0) | 56.7 (82.2) | 40.5 (53.6) |
| Number of staff assisting, n* | | | | | | | | |
| 0 | 31 (2.4%) | 0 | 0 | 0 | 4 (1.9%) | 1 (0.3%) | 4 (5.1%) | 22 (9.6%) |
| 1 | 406 (31.6%) | 4 (1.5%) | 26 (30.2%) | 14 (12.3%) | 135 (62.8%) | 72 (24.8%) | 30 (38.5%) | 125 (54.3%) |
| 2 | 732 (57.0%) | 223 (82.2%) | 57 (66.3%) | 74 (64.9%) | 72 (33.5%) | 192 (66.2%) | 36 (46.2%) | 78 (33.9%) |
| 3 | 103 (8.0%) | 40 (14.8%) | 3 (3.5%) | 22 (19.3%) | 4 (1.9%) | 24 (8.3%) | 6 (7.6%) | 4 (1.7%) |
| 4 | 12 (0.9%) | 4 (1.5%) | 0 | 4 (3.5%) | 0 | 1 (0.3%) | 2 (2.6%) | 1 (0.4%) |

*Difference between the seven types of surgery=p<0.001.
Abd, abdominal; Q, quartile; Reg, registered.

**Table 3** ORs for mobilisation within 6 and 24 hours after abdominal or cardiothoracic surgery

| Variable | Value | Mobilisation within 6hours | | | Mobilisation within 24 hours | | |
|---|---|---|---|---|---|---|---|
| | | n (%) of event | OR (95% CI) | P value | n (%) of event | OR (95% CI) | P value |
| Category of surgery | Reference: Cardiac | 120 (41.1%) | 1.00 | <0.0001* | 279 (95.5%) | 1.00 | 0.12* |
| | Thoracic | 55 (59.8%) | 2.13 (1.32 to 3.43) | 0.0019 | 91 (97.8%) | 2.12 (0.47 to 9.57) | 0.33 |
| | Major upper Abd | 34 (27.9%) | 0.55 (0.35 to 0.88) | 0.012 | 118 (95.2%) | 0.92 (0.34 to 2.47) | 0.86 |
| | Minor upper Abd | 160 (66.1%) | 2.80 (1.96 to 3.98) | <0.0001 | 234 (94.4%) | 0.78 (0.36 to 1.69) | 0.53 |
| | Intestinal | 179 (54.1%) | 1.69 (1.23 to 2.32) | 0.0012 | 317 (94.3%) | 0.78 (0.38 to 1.60) | 0.50 |
| | Major lower Abd | 43 (48.3%) | 1.34 (0.83 to 2.16) | 0.23 | 91 (98.9%) | 4.24 (0.55 to 32.86) | 0.17 |
| | Minor lower Abd | 147 (55.7%) | 1.80 (1.29 to 2.52) | 0.0006 | 265 (98.5%) | 3.09 (0.99 to 9.59) | 0.051 |
| Procedure | Reference: Elective | 620 (55.4%) | 1.00 | <0.0001* | 1119 (98.5%) | 1.00 | <0.0001* |
| | Acute | 62 (33.0%) | 0.40 (0.29 to 0.55) | <0.0001 | 155 (81.6%) | 0.07 (0.04 to 0.12) | <0.0001 |
| | Subacute† | 52 (44.4%) | 0.64 (0.44 to 0.94) | 0.024 | 112 (94.1%) | 0.24 (0.10 to 0.60) | 0.0021 |
| Duration of anaesthesia | Reference: 0–4 hours | 385 (58.0%) | 0.62 (0.50 to 0.76) | <0.0001 | 651 (96.7%) | 0.71 (0.41 to 1.22) | 0.22 |
| | >4 | 349 (46.0%) | | | 736 (95.5%) | | |
| Arrival in postoperative care | Reference: Day | 404 (61.2%) | 1.00 | <0.0001* | 652 (97.8%) | 1.00 | <0.0001* |
| | Evening | 323 (47.1%) | 0.56 (0.45 to 0.70) | <0.0001 | 676 (96.8%) | 0.71 (0.36 to 1.37) | 0.31 |
| | Night | 11 (12.8%) | 0.09 (0.05 to 0.18) | <0.0001 | 67 (75.3%) | 0.07 (0.03 to 0.14) | <0.0001 |
| Perioperative bleeding, mL | Reference: <999 mL | 509 (49.9%) | 0.48 (0.31 to 0.75) | 0.0012 | 999 (96.3%) | 0.58 (0.24 to 1.40) | 0.22 |
| | ≥1000 mL | 31 (32.3%) | | | 91 (93.8%) | | |
| Epidural anaesthesia | Reference: No | 606 (53.0%) | 0.68 (0.52 to 0.89) | 0.0051 | 1115 (96.0%) | 0.82 (0.44 to 1.54) | 0.54 |
| | Yes | 117 (43.5%) | | | 259 (95.2%) | | |
| Spinal anaesthesia | Reference: No | 653 (51.4%) | 0.97 (0.69 to 1.37) | 0.87 | 1233 (95.7%) | 2.24 (0.69 to 7.25) | 0.18 |
| | Yes | 75 (50.7%) | | | 148 (98.0%) | | |
| Sex | Reference: Male | 373 (49.3%) | 1.20 (0.97 to 1.47) | 0.093 | 732 (95.6%) | 1.22 (0.72 to 2.07) | 0.45 |
| | Female | 361 (53.8%) | | | 658 (96.3%) | | |
| Age, years | Reference: 18–69 | 461 (54.0%) | 0.78 (0.63 to 0.96) | 0.021 | 842 (97.1%) | 0.48 (0.28 to 0.82) | 0.0068 |
| | ≥70 | 277 (47.8%) | | | 553 (94.2%) | | |
| Obesity | Reference: BMI<30 | 534 (50.8%) | 1.17 (0.92 to 1.49) | 0.20 | 1026 (96.2%) | 1.14 (0.59 to 2.20) | 0.69 |
| | BMI>30 | 195 (54.8%) | | | 351 (96.7%) | | |
| Functional status | Reference: Independent | 735 (51.9%) | 0.20 (0.06 to 0.069) | 0.011 | 1382 (96.2%) | 0.10 (0.03 to 0.30) | <0.0001 |
| | Dependant | 3 (17.6%) | | | 13 (72.2%) | | |
| Received preoperative information | Reference: No | 392 (52.3%) | 0.90 (0.73 to 1.11) | 0.32 | 715 (93.7%) | 4.03 (2.08 to 7.83) | <0.0001 |
| | Yes | 330 (49.7%) | | | 661 (98.4%) | | |

Continued

**Table 3** Continued

| Variable | Value | Mobilisation within 6 hours | | | Mobilisation within 24 hours | | |
|---|---|---|---|---|---|---|---|
| | | n (%) of event | OR (95% CI) | P value | n (%) of event | OR (95% CI) | P value |
| Type of hospital | Reference university | 448 (47.6%) | 1.00 | <0.0001* | 919 (96.0%) | 1.00 | 0.33* |
| | Regional county | 232 (57.1%) | 1.47 (1.16 to 1.86) | 0.0013 | 393 (95.2%) | 0.81 (0.47 to 1.41) | 0.46 |
| | Local county | 58 (69.0%) | 2.46 (1.52 to 3.97) | 0.0002 | 83 (98.8%) | 3.43 (0.47 to 25.31) | 0.23 |
| Type of ward | Reference: IMU | 23 (60.5%) | 1.00 | 0.0005* | | | |
| | Postop/ICU | 468 (57.8%) | 0.89 (0.46 to 1.74) | 0.74 | | | |
| | Surgical ward | 240 (47.0%) | 0.58 (0.29 to 1.13) | 0.11 | | | |

*P value for the entire effect/factor/variable.
†Acute surgery was defined as surgery within 24 hours after decision of surgery and subacute surgery within a week after the decision.
Abd, abdominal; ICU, intensive care unit; IMU, intermediate unit; Postop, postoperative unit.

associated with the longest time to mobilisation (mean 11 hours 03 min, SD 4:08 hours), whereas minor upper abdominal procedures were associated with the shortest time (mean 5 hours 07 min, SD 5:38 hours). Online supplemental figure 1 shows the times that patients arrived at the postoperative care units during the day and night and when the first mobilisation was started. Most patients arrived at the postoperative wards between 10:30 and 17:00. Most of the first mobilisations started around 8 and between 14:00 and 20:00.

In total, 59 patients (4%) were not mobilised within the first 24 hours of surgery, and stated reasons (several reasons could be listed) were: circulatory instability (n=7), respiratory instability (n=15), sedation (n=12), intubation (n=6), pain (n=9), nausea (n=2), tiredness (n=7) or not considered a priority by the healthcare professionals (n=3). The reasons were similar across seven categories of surgery (data not shown).

### Time out of bed and type of mobilisation

The mean duration of time out of bed was 27 min (SD 1 hour 16 min). Patients who had undergone cardiac surgery, intestinal surgery and major lower abdominal surgery had the shortest time out of bed during their first mobilisation (table 2).

Except that all patients were mobilised sat on the edge of the bed, 76% stood up by the bed, 15% sat in a chair and 22% walked around the first time they were out of bed. There were significant differences between the groups of surgical categories in all activities (p<0.001). When comparing the results between the seven types of surgery, the group that had cardiac surgery was the one that most often also were mobilised to standing up by the bed (89%), those that had lung surgery most often sat in a chair (82%) and those that underwent minor upper abdominal surgery were most often up and walked around (18%) (p<0.001). Commonly, two from the staff were assisting during the first mobilisation (table 2). Most often, a nurse (in 60% of the mobilisations) or an assistant nurse (also in 60% of the mobilisations) assisted the patients during the mobilisation.

### Factors associated with mobilisation

The results of the OR for mobilisation within 6 and 24 hours are presented in table 3, and for 3 and 12 hours, see online supplemental table 2. Minor upper abdominal surgery had the highest odds for mobilisation during the first 6 hours postoperatively (OR: 2.80, 95% CI: 1.96 to 3.98). Decreased odds were factors as acute surgery (OR: 0.40, 95% CI: 0.29 to 0.55), arrival at the postoperative care unit during evenings (OR: 0.56, 95% CI: 0.45 to 0.70) and nights (OR: 0.09, 95% CI: 0.05 to 0.18), duration of anaesthesia>4 hours (OR: 0.62, 95% CI: 0.50 to 0.76) and bleeding>999 mL (OR: 0.48, 95% CI: 0.31 to 0.75). The factors associated with increased odds of mobilisation even earlier (within 3 hours after surgery) were minor upper abdominal surgery (OR: 2.20, 95% CI:

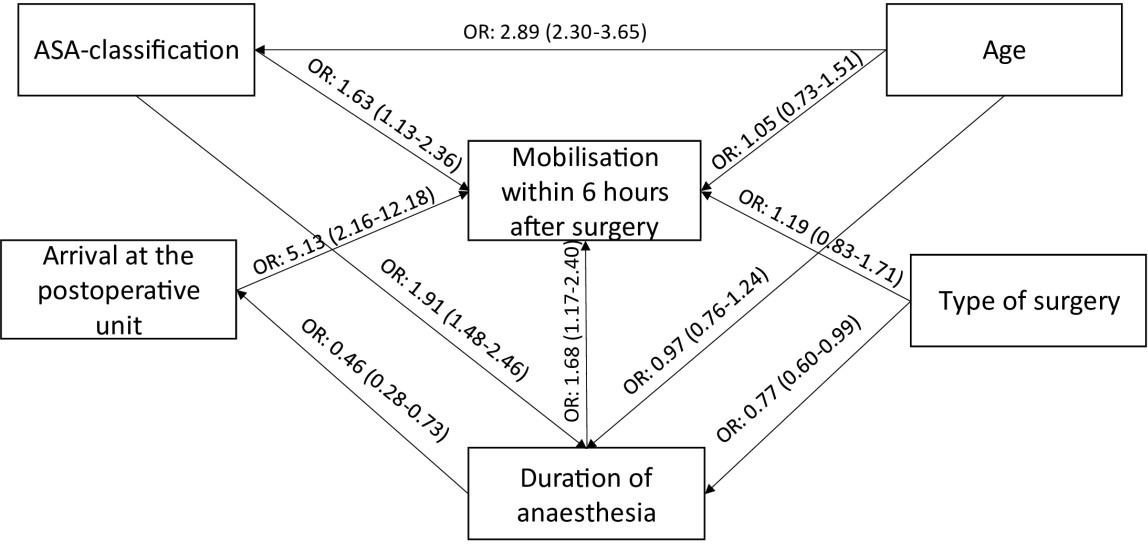

**Figure 2** General structural equation model analysis displaying correlations between time to first mobilisation, categorised as within 6 hours and the factors; ASA classification; age; arrival at the postoperative recovery unit; type of surgery; duration of surgery. ASA, American Society of Anaesthesiologists.

1.48 to 3.26) and having surgery at a local county hospital (OR: 2.24, 95% CI: 1.36 to 3.69).

The strongest factors associated with increased odds of the first mobilisation within 24 hours were lower abdominal surgery (OR: 4.24, 95% CI: 0.55 to 32.86) (major surgery) and minor surgery (OR: 3.09, 95% CI: 0.99 to 9.59), receiving preoperative information by the physiotherapist (OR: 4.03, 95% CI: 2.08 to 7.83) and surgery performed in local county hospitals (OR: 3.43, 95% CI: 0.47 to 25.31). Factors that decreased the odds were acute surgery and arriving to postoperative care during the night (OR: both 0.07).

According to the results of the GSEM (figure 2), the following of the included factors were found to increase the probability of being mobilised within 6 hours of surgery (conditional on other factors): an ASA classification of 1 and 2 (OR: 1.63, 95% CI: 1.13 to 2.36), arriving at the postoperative recovery unit in the daytime/afternoon (OR: 5.13, 95% CI: 2.16 to 12.18) and a duration of anaesthesia of less than 4 hours (OR: 1.68, 95% CI: 1.17 to 2.40).

## DISCUSSION
This is the first prospective study describing mobilisation routines after cardiothoracic and abdominal surgery. In this national, multicentre, observational study, which included a total of 1492 patients, it was found that most patients (96%) were mobilised within the first 24 hours, whereas more than half of the cohort (52%) were

mobilised within 6 hours. In addition to sitting on the edge of the bed, 76% stood up during the first mobilisation. Arriving at the postoperative recovery unit in the daytime, a duration of anaesthesia of less than 4 hours and having few comorbidities (ASA 1 or 2) were independently associated with being mobilised within 6 hours of surgery.

Even though almost all patients were mobilised within 24 hours, there were some differences regarding the time to first mobilisation between the different types of surgery. Possible explanations are diverse methods of analgesia and different inflammatory responses after the surgical trauma[20] due to different proportions of a malign cause of surgery where cancer is the most common cause of surgery for major upper abdominal surgery in contrary to cardiac surgery. The duration of first mobilisation also differed between the surgery types but not to the same extent as the time to mobilisation. The average time out of bed was 27 min, but was about twice that after lung surgery. A shorter duration of anaesthesia during the pulmonary procedures via thoracotomy may be one explanatory factor compared with cardiac surgery via sternotomy or larger abdominal incision with longer duration of anaesthesia.

A more upright position appears to be essential to normalise functional residual capacity and respiratory function after anaesthesia and surgery.[21–23] In addition, the upright position may also help to reinflate the lung after atelectasis.[12 13 24] An earlier and more intensive

mobilisation, that is, to standing and walking, may therefore be preferable, especially for those at high risk of postoperative complications. In the current study, all mobilised patients sat on the edge of the bed, but a majority also stood up by the bed and 21% even walked around. This is encouraging since the results from other studies show that compliance with Enhanced Recovery After Surgery (ERAS) recommendations is relatively poor with only 25%–50% reaching the mobilisation targets after colorectal surgery.[25 26] Nevertheless, there are few absolute contraindications for mobilisation, and as including more active approaches in clinical pathways has been proven to reduce complications and accelerate recovery,[25 26] there may be room for improvement to fulfil the recommendations concerning mobilisation.[7]

The findings of the present study indicate that arriving at the postoperative recovery unit in the daytime/afternoon is the strongest predictor of an earlier mobilisation (within 6 hours after surgery) as is a low ASA classification and a duration of anaesthesia of less than 4 hours. The reason why arrival during daytime is such a strong predictor is not known even if it is possible to speculate about variations in number of staff and differences in composition of professions during the day and night. Future studies are needed on this topic. However, what is known is that early mobilisation, including sitting in a chair, standing and walking is not only feasible, but also leads to improved oxygenation as $SpO_2$ and $PaO_2$ in patients after pulmonary lobectomy[22] and different abdominal surgery.[14 15 27] Thus, to facilitate early mobilisation after surgery, the focus might be to identify and target patients with comorbidities (ASA classification≥2) as it has been found that an ASA class≥2 is associated with an increased risk of postoperative pulmonary complications.[28] In current trial, we did not investigate PPC, hence we do not know if patients with an early mobilisation developed fewer PPC compared with those with a late mobilisation, or whether mobilisation itself constitutes a protective factor for PPC. Future trials are needed to explore that. Furthermore, to facilitate mobilisation it may be necessary to allocate all different healthcare professionals in the team at the postoperative recovery unit also to the evenings, when the patients with a prolonged surgery arrive.[29 30]

In 1958, Powers[31] suggested that 'early mobilisation' should occur on the first day after hernioplasty and 'prompt mobilisation' should occur on the day of surgery. However, surgery as well as preoperative, perioperative and postoperative care have developed considerably since then. Despite the frequent use of 'early', there is still no consensus about its definition.[2 7] In the current trial, national practice after different surgeries was investigated. The results indicate that more than half of the patients were mobilised within 6 hours. It remains to be evaluated whether mobilisation within 6 hours after surgery may be used as a definition of early mobilisation in today's practice. But to increase transparency and avoid misuse and misinterpretation of the term 'early mobilisation', it is high time to discuss and come to a consensus about a definition.

## Strengths and limitations

The prospective, multicentre design of this study, including patients from all parts of Sweden, and the relatively high participation rate counteract selection bias. To determine the standard practice of postoperative mobilisation, we aimed to include at least 700 patients, leaving approximately 100 patients in each surgical category on which we could perform subgroup analyses. The large study sample enabled the planned analyses of the different surgical categories and exploration of related variables. Still, approximately 20% of the included patients had no registered outcome data. Outcomes were determined by observations and relied on correct documentation by investigators. There may be several reasons for the missing data, such as lack of staff, problems reaching all staff with information about the study and logistic challenges. Another limitation was that no patients were included during weekends, and therefore the results may not completely reflect common practice. Adverse events were not registered during the mobilisation. These would have given valuable information concerning the association between risk and when the mobilisation was performed. In addition, no score of frailty was included among the variables connected to mobilisation praxis but it is factor which could have had impact on the outcome. Future trials may also include a frailty index to further explore the subject.

## CONCLUSION

Most of the patients in Sweden undergoing abdominal and cardiothoracic surgery that involved anaesthesia exceeding 2 hours were mobilised within the first 24 postoperative hours, with more than half mobilised within 6 hours, and the first mobilisation included standing in the majority of cases. Factors associated with earlier mobilisation were daytime arrival at the postoperative recovery unit, shorter duration of anaesthesia and few comorbidities (low ASA classification). This overview of the national practice reveals unique baseline information to which other countries may be compared.

**Author affiliations**
[1]Department of Health and Rehabilitation/Physiotherapy, Institute of Neuroscience and Physiology and Department of Surgery, Institute of Clinical Sciences, Sahlgrenska Academy, University of Gothenburg, Goteborg, Sweden
[2]Department of Community Medicine and Rehabilitation, Physiotherapy, Umeå University, Umeå, Sweden
[3]University Health Care Research Center and Department of Cardiothoracic and Vascular Surgery, Faculty of Medicine and Health, Örebro University, Örebro, Sweden
[4]Department of Anaesthesiology and Intensive Care, Södersjukhuset, Stockholm, Sweden/ Department of Clinical Science and Education, Karolinska Institutet, Södersjukhuset, Sweden
[5]Department of Anaesthesiology and Intensive Care, Institute of Clinical Sciences, Sahlgrenska Academy, Gothenburg University, Gothenburg, Sweden/ Department of Anaesthesiology and Intensive Care, Sahlgrenska University Hospital, Gothenburg, Sweden

⁶Department of Neurobiology, Care Sciences and Society, Division of Physiotherapy, Karolinska Institutet, Stockholm, Sweden/ Women's Health and Allied Health Professionals Theme, Medical Unit Occupational Therapy and Physiotherapy, Karolinska University Hospital, Stockholm, Sweden

**Contributors** MFO had full access to all the data in the study and was responsible for the integrity of the data and the accuracy of the data analysis. Concept and design: MFO, MS, EW, ARS, LB, MN-B and AS-R. Acquisition and analysis: MFO and AS-R. Interpretation of results: MFO, MS, EW, ARS, LB, MN-B and AS-R. Critical revision of the manuscript: MFO, MS, EW, ARS, LB, MN-B and AS-R. Statistical analysis: Statistical consultants, MFO and AS-R. Guarantor: MFO.

**Funding** This study was financed by grants from the Swedish state under the agreement between the Swedish government and the county councils, the ALF-agreement (ALFGBG-965563), by The Local Research and Development Council, Gothenburg and Södra Bohuslän (VGFOUGSB-970106) and by Nyckelfonden, Örebro University Hospital Research Foundation (Grant number N/A).

**Competing interests** None declared.

**Patient and public involvement** Patients and/or the public were not involved in the design, or conduct, or reporting, or dissemination plans of this research.

**Patient consent for publication** Not applicable.

**Ethics approval** This study involves human participants. The Swedish Ethical Review Authority approved the study (registration number 2020-03108). Because of the observational study design of current practice, patient consent was waived.

**Provenance and peer review** Not commissioned; externally peer reviewed.

**Data availability statement** Data are available on reasonable request. Data, materials and analytical methods will be shared on reasonable request.

**ORCID iDs**
Monika Fagevik Olsén http://orcid.org/0000-0002-0207-4105
Anna Svensson-Raskh http://orcid.org/0000-0002-7949-5864

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
