## [Reviewer comments · BMJ Open]

ARTICLE DETAILS

TITLE (PROVISIONAL)	First mobilisation after abdominal and cardiothoracic surgery - When is it actually performed? A national, multicentre, cross-sectional study
AUTHORS	Fagevik Olsén, Monika; Sehlin, Maria; Westerdahl, Elisabeth; Schandl, Anna; Block, Linda; Nygren-Bonnier, Malin; Svensson-Raskh, Anna

VERSION 1 – REVIEW

REVIEWER	Battle, Ceri Morrison Hospital, Welsh Institute of Biomedical and Emergency Medicine Research
REVIEW RETURNED	28-Nov-2023

GENERAL COMMENTS	The authors have completed a national, multicentre, cross-sectional study that explores mobilisation practices following abdominal and cardiothoracic surgery. This is an important study as it advances knowledge regarding current practice and identifies factors associated with early mobilisation. This study has been very well conducted and this is an excellent manuscript. As a result, I have very few comments for the authors. Abstract: Clear and informative Background: Well written and includes relevant current literature. Provides the reader with clear justification for the study. Aims clearly identified. Methods: Well written and provides sufficient information according to STROBE guidance. Could you include a statement about how missing data was handled in the methods please. Results: Well-reported and easily interpreted. Tables, figures and supplementary material all very well presented. Discussion: Again - very well written and covers relevant discussion points raised in the results. Conclusions, strengths and limitations are all appropriate. Did you collect any data regarding frailty? I think this is such a hot topic at present in rehabilitation research and wider contexts, it would be useful to the reader. If not, perhaps this could be identified as a potential limitation, or area for future studies to consider.
---

	I would like to congratulate the authors on this work. It has been very well conducted and well presented in this manuscript. As a result, the work was a pleasure to review. Thank you.
--	--

REVIEWER	Lee, A Monash University
-----------------	-----------------------------

REVIEW RETURNED	11-Dec-2023
-------------

GENERAL COMMENTS	This study is outlining the clinical practice with regards to mobilisation in upper abdominal and cardiothoracic surgery. Its aims are clearly outlined and the study provides valuable insight into clinical practice. However, some points need clarification or additional detail and are detailed below. Abstract: Page 2, line 36, it is not clear what at lite bed edge sitting is referring to? Presumably this is sitting on the edge of the bed, but could this be phrased a little more clearly in the classification? What is the time frame for 76% standing by the bed and 22% away from the bedside? Methods: Page 6, line 15, was the staff member screening patients the same person who screened all patients at each site? Presumably, there was one designated person for this role at each site. Was this staff member a physiotherapy or nurse and what was their experience in surgical practice? It is noted that patients with minor abdominal (including laparoscopic gynaecological procedures) were included. Given that these categories could be considered low risk for postoperative pulmonary complications compared to other procedures, what was the rationale for including patients within this study? Page 7, line 44, any reason why min-max has been selected for a measure of variance of median rather than interquartile range? In the statistical analysis, it is mentioned that differences between groups will be analysed, but it is not clear exactly which groups will be compared? Is that according to surgery type? Results: Table 1 – how was functional status of partially or totally dependent defined? Was this determined subjectively or using a scale? Table 1 – Age could be rounded up to 0 decimal places. For duration of anaesthesia and duration of surgery, with the units in hours, the decimal points are confusing to read. For example, for all patients, duration of anaesthesia is 4 hours and 60 mins? But wouldn't this be 5 hours? Could the authors clarify how the units work for these measures. Same point in Table 2 for time to first mobilisation and duration of mobilisation. Any reason why both mean (SD) and median (IQR) have both been reported? What was the distribution of this data? For Table 1, 2, there is a lack of a legend defining all the abbreviations used in the tables, these should be added. Page 9, line 47, According to Table 2, it appears that those with major lower abdominal surgery had the shortest time out of bed during their first mobilisation rather than the minor lower abdominal surgery It is not clear in Table 2 where the information regarding the assistance with mobilisation is stated?
--

	Comparisons between those who had undergone cardiac surgery who had stood by the bed compared to pulmonary surgery sitting in a chair or walking, but it is not clear where this data is originating from within the Table. The data on actual activity is lacking and would be very helpful to have this included. Some information is missing in Figure 1 with regards to some categories of surgery as the included number of patients is short of 1492. For the factors associated with mobilisation, it would be useful to see a little more interpretation with regards to the odds ratio. For example, on page 10, line 21, it is stated that minor upper abdominal surgery had the highest odds for mobilisation with the first 6 hours, but reporting this in the context of the value of the odds ratio would be useful. Some commentary related to the information presented for mobilisation within 24 hours in the text would be useful to include. Are there any key messages with regards to the information related to mobilisation within 3 hours or 12 hours to comment on? What was the definition of acute vs subacute surgery? Discussion Page 11, line 29 – what do the authors mean by a ‘different proportions of a malign cause of surgery’? IN the second paragraph, there is reference to a Danish study identifying sternotomy and thoracoabdominal incisions as risk factors for PPC, but it is not clear the link between the information presented and the detail for time to first mobilisation? Could the authors clarify this paragraph? The third paragraph refers to the benefits of upright positioning, but the sub-detail of upright with regards to sitting on the bed edge, standing and mobilising doesn’t appear to be included in the manuscript. This detail would be helpful (assuming it is currently including under a general heading of mobilisation in Table 2)? There is a reference to ERAS recommendations; could the authors specify what surgical categories these guidelines are referring to? Are they all types of surgery that were included in this study? On page 12, line 17, the authors are referring to room for improvement, presumably in the field of greater encouragement for mobilisation following surgery in each of these categories? There is a mention that more active approaches in clinical pathways has reduced complications and accelerated recovery – some reference to support this point would be useful. In the discussion with reference to the predictor of early mobilisation, it should be stated that this refers to mobilisation within 6 hours. Page 12, line 38 – when referring to targeting patients with higher ASA classifications etc, a reference to targeting them for early mobilisation/intervention may be useful. The comment on sufficient personnel to assist in mobilisation is difficult to interpret in the absence of clear data related to this point. The following sentence is not clear: ‘To facilitate mobilisation not only during the daytime but it is also of great importance to have sufficient number of healthcare personnel who can assist in the procedures.’ Was Powel’s referring to all types of surgery in 1958? Minor points: Page 4, line 9, a word ‘of’ is missing before pre, peri and postoperative care, just to make the phrase a little clearer.
--	---

	Page 4, line 29 – secretion should read ‘secretions’ Page 5, line 15, needs a , after hospitals, see Supplementary file Table A. Page 7, line 12. Where you have stated the secondary outcomes, this sentence could be clarified to read: ‘Secondary outcomes were the type and duration of the first mobilisation, number of staff assisting and to identify which factors were associated with early mobilisation, defined as within six hours of surgery.’ Page 7, line 31 – rephrase to be ‘until 24 hours from admission to a postoperative unit’ Page 8, line 14, select a different word to ‘thereafter’. Suggest rephrasing to ‘following this...’ Page 9, line 52 – rephrase “Besides that all patients who were mobilised sat on the bed edge...” This is a little unclear with the sentence starting with ‘Besides’ Page 12, line 28, the word ‘and’ should be stated after duration of anaesthesia >- 4 hours are associated....’
--	---

VERSION 1 – AUTHOR RESPONSE

Referee 1

Comments to the Author:

The authors have completed a national, multicentre, cross-sectional study that explores mobilisation practices following abdominal and cardiothoracic surgery. This is an important study as it advances knowledge regarding current practice and identifies factors associated with early mobilisation. This study has been very well conducted and this is an excellent manuscript. As a result, I have very few comments for the authors.

Abstract:

Clear and informative

Background:

Well written and includes relevant current literature. Provides the reader with clear justification for the study. Aims clearly identified.

Methods:

Well written and provides sufficient information according to STROBE guidance.

Could you include a statement about how missing data was handled in the methods please.

Results:

Well-reported and easily interpreted. Tables, figures and supplementary material all very well presented.

Discussion:

Again - very well written and covers relevant discussion points raised in the results. Conclusions, strengths and limitations are all appropriate.

Did you collect any data regarding frailty? I think this is such a hot topic at present in rehabilitation research and wider contexts, it would be useful to the reader. If not, perhaps this could be identified as a potential limitation, or area for future studies to consider.

I would like to congratulate the authors on this work. It has been very well conducted and well presented in this manuscript. As a result, the work was a pleasure to review. Thank you.

Comments from the authors:

Thank you for revising our manuscript and for the positive response. The two things you pointed out which we have missed to include in the text are now included, the first one in the methods and the second one in the end of “strengths and limitations”.

Referee 2

This study is outlining the clinical practice with regards to mobilisation in upper abdominal and cardiothoracic surgery. Its aims are clearly outlined and the study provides valuable insight into clinical practice.

Authors’ reply: Thank you for reviewing our manuscript and for your comment that the content is of importance. We have considered your suggestions for improvements of our manuscript and have made changes to the text as described after each of your comments below:

Some points need clarification or additional detail and are detailed below.

Abstract:

Page 2, line 36, it is not clear what at lite bed edge sitting is referring to? Presumably this is sitting on the edge of the bed, but could this be phrased a little more clearly in the classification?

Authors’ reply: Sorry, this was an error in the text. It should have been “to at least bed edge sitting”. This is now corrected.

What is the time frame for 76% standing by the bed and 22% away from the bedside?

Authors’ reply: The timeframe is the same as both activities were performed during the first mobilization the patients did after the surgery. We have included some extra word in the abstract to make it clearer.

Methods:

Page 6, line 15, was the staff member screening patients the same person who screened all patients at each site? Presumably, there was one designated person for this role at each site. Was this staff member a physiotherapy or nurse and what was their experience in surgical practice?

Authors’ reply: We agree that information about this was missing and it is now included in the text.

It is noted that patients with minor abdominal (including laparoscopic gynaecological procedures) were included. Given that these categories could be considered low risk for postoperative pulmonary complications compared to other procedures, what was the rationale for including patients within this study?

Authors’ reply: We agree that women undergoing laparoscopic gynaecological procedures are in lower risk for postoperative pulmonary complications. However, as we wanted to present a broad perspective on clinical practice concerning mobilisation in this manuscript, we included also this category of surgery in the study.

Page 7, line 44, any reason why min-max has been selected for a measure of variance of median rather than interquartile range?

Authors’ reply: The data is given as interquartile range and the text is revised under “statistics”

In the statistical analysis, it is mentioned that differences between groups will be analysed, but it is not clear exactly which groups will be compared? Is that according to surgery type?

Authors' reply: Yes, we mean the different types of surgery and we have now revised the sentence to make it clearer.

Results:

Table 1 – how was functional status of partially or totally dependent defined? Was this determined subjectively or using a scale?

Authors' reply: Functional status is preoperatively assessed according to each hospital's clinical practice and recorded in the patient files. No national scale was used. We have included information about it in the text.

Table 1 – Age could be rounded up to 0 decimal places. For duration of anaesthesia and duration of surgery, with the units in hours, the decimal points are confusing to read. For example, for all patients, duration of anaesthesia is 4 hours and 60 mins? But wouldn't this be 5 hours? Could the authors clarify how the units work for these measures. Same point in Table 2 for time to first mobilisation and duration of mobilisation.

Authors' reply: We have deleted the decimals in the table with age. In the reporting of duration of anaesthesia and surgery and time to first mobilisation decimal numbers and not hours/minutes are given. This can be confusing. We have revised the text in the results to make it clearer.

Any reason why both mean (SD) and median (IQR) have both been reported? What was the distribution of this data?

Authors' reply: We included both after discussion with our statistician. The data is skew with outliers which can be seen when comparing mean and median.

For Table 1, 2, there is a lack of a legend defining all the abbreviations used in the tables, these should be added.

Authors' reply: When inserted Table 2 in the format of the journal an error was made. The last part of the table and the abbreviations were never included. The information is now included.

Page 9, line 47, According to Table 2, it appears that those with major lower abdominal surgery had the shortest time out of bed during their first mobilisation rather than the minor lower abdominal surgery.

Authors' reply: We are so grateful that you saw this error. It should have been major lower abdominal surgery and the text is now corrected.

It is not clear in Table 2 where the information regarding the assistance with mobilisation is stated? Comparisons between those who had undergone cardiac surgery who had stood by the bed compared to pulmonary surgery sitting in a chair or walking, but it is not clear where this data is originating from within the Table. The data on actual activity is lacking and would be very helpful to have this included.

Authors' reply: See previous reply. The data concerning what kind of mobilisation, which was performed, and number of staff included is now to be found in Table 2. In addition, the table is revised according to the journal's instructions.

Some information is missing in Figure 1 with regards to some categories of surgery as the included number of patients is short of 1492.

Authors' reply: Sorry, the box with the information had been reduced so all content was not visible. We have now corrected it.

For the factors associated with mobilisation, it would be useful to see a little more interpretation with regards to the odds ratio. For example, on page 10, line 21, it is stated that minor upper abdominal surgery had the highest odds for mobilisation with the first 6 hours, but reporting this in the context of the value of the odds ratio would be useful.

Some commentary related to the information presented for mobilisation within 24 hours in the text would be useful to include. Are there any key messages with regards to the information related to mobilisation within 3 hours or 12 hours to comment on?

Authors' reply: We have included some additional text concerning the odds to be mobilised within the first 3 and 24 hours.

What was the definition of acute vs subacute surgery?

Authors' reply: Acute surgery was defined as surgery within 24 hours after decision of surgery and subacute surgery within a week after the decision. This is now included after both tables.

Discussion

Page 11, line 29 – what do the authors mean by a 'different proportions of a malign cause of surgery'?

Authors' reply: There are different causes of the seven types of surgery. Most of the patients undergoing major upper abdominal surgery have a cancer diagnose which is not the case for patients undergoing cardiac surgery. We have added some text to, hopefully, make it more understandable.

In the second paragraph, there is reference to a Danish study identifying sternotomy and thoracoabdominal incisions as risk factors for PPC, but it is not clear the link between the information presented and the detail for time to first mobilisation? Could the authors clarify this paragraph?

Authors' reply: We agree that the link was weak and have removed the statement and reference.

The third paragraph refers to the benefits of upright positioning, but the sub-detail of upright with regards to sitting on the bed edge, standing and mobilising doesn't appear to be included in the manuscript. This detail would be helpful (assuming it is currently including under a general heading of mobilisation in Table 2)?

Authors' reply: See above. The detailed information is now to be found in Table 2.

There is a reference to ERAS recommendations; could the authors specify what surgical categories these guidelines are referring to?

Are they all types of surgery that were included in this study?

Authors' reply: The guidelines refer to colorectal surgery and we have now included it in the paragraph. Some of the patients in our study were followed an ERAS program and, later on, we will write a specific manuscript about this and the ratio of patients who fulfil the recommendations.

On page 12, line 17, the authors are referring to room for improvement, presumably in the field of greater encouragement for mobilisation following surgery in each of these categories)

Authors' reply: Here, we refer to reference 7 but we agree that the message was not clear and have added some words, therefore.

There is a mention that more active approaches in clinical pathways have reduced complications and accelerated recovery – some reference to support this point would be useful.

Authors' reply: We agree and have added references.
In the discussion with reference to the predictor of early mobilisation, it should be stated that this refers to mobilisation within 6 hours.

Authors' reply: It is now included.

Page 12, line 38 – when referring to targeting patients with higher ASA classifications etc, a reference to targeting them for early mobilisation/intervention may be useful.

Authors' reply: The text has been revised (see also the comment below) and a reference is added.

The comment on sufficient personnel to assist in mobilisation is difficult to interpret in the absence of clear data related to this point.

Authors' reply: The data is now given in Table 2.

The following sentence is not clear: 'To facilitate mobilisation not only during the daytime but it is also of great importance to have sufficient number of healthcare personnel who can assist in the procedures.'

Authors' reply: We have rewritten the paragraph and hope that the message now is clearer.

Was Powel's referring to all types of surgery in 1958?

Authors' reply: In the study by Powels patients undergoing hernioplasty were included. We have therefore included this information in the text.

Minor points:

Page 4, line 9, a word 'of' is missing before pre, peri and postoperative care, just to make the phrase a little clearer.

Page 4, line 29 – secretion should read 'secretions'

Page 5, line 15, needs a , after hospitals, see Supplementary file Table A.

Page 7, line 12. Where you have stated the secondary outcomes, this sentence could be clarified to read: 'Secondary outcomes were the type and duration of the first mobilisation, number of staff assisting and to identify which factors were associated with early mobilisation, defined as within six hours of surgery.'

Page 7, line 31 – rephrase to be 'until 24 hours from admission to a postoperative unit'

Page 8, line 14, select a different word to 'thereafter'. Suggest rephrasing to 'following this...'

Page 9, line 52 – rephrase "Besides that all patients who were mobilised sat on the bed edge..." This is a little unclear with the sentence starting with 'Besides'

Page 12, line 28, the word 'and' should be stated after duration of anaesthesia >- 4 hours are associated....'

Authors' reply: Thank you for your additional comments and we have changed the text accordingly.

Specific editorial comments:

- Please limit each bullet point of the Strengths and Limitations section (after the abstract) to a single sentence.

Authors: This is now changed

- Please include the patient and public involvement statement under its own heading within the Methods section.

Authors: We have inserted a specific sub-heading in the text

- Please clarify why this study is registered as an interventional study NCT04729634 and entitled a 'survey'.

Authors: When we named the study, we used the word "Survey" to describe that we wanted to map or chart the clinical practice. We have since then understood that the word may be interpreted differently. We therefore now call it an observational study.

In addition, we have revised the tables as we received information that the tables could not include more than 9 columns.

VERSION 2 – REVIEW

REVIEWER	Battle, Ceri Morrison Hospital, Welsh Institute of Biomedical and Emergency Medicine Research
REVIEW RETURNED	22-Jan-2024
GENERAL COMMENTS	Thank you for revising your manuscript in response to the review. I have nothing further to add and would again like to thank the authors for their work.